# CXCR1- or CXCR2-modified CAR T cells co-opt IL-8 for maximal antitumor efficacy in solid tumors

Linchun Jin [1], Haipeng Tao[2], Aida Karachi[1], Yu Long[1,2], Alicia Y. Hou[1], Meng Na[2], Kyle A. Dyson [1], Adam J. Grippin [1], Loic P. Deleyrolle[1,3], Wang Zhang[2], Didier A. Rajon[1,3], Qiong J. Wang[4,6], James C. Yang [4], Jesse L. Kresak[5], Elias J. Sayour[1,3], Maryam Rahman[1,3], Frank J. Bova[1,3], Zhiguo Lin[2], Duane A. Mitchell[1,3] & Jianping Huang [1,3]

Chimeric antigen receptor (CAR) T-cell therapy targeting solid tumors has stagnated as a result of tumor heterogeneity, immunosuppressive microenvironments, and inadequate intratumoral T cell trafficking and persistence. Early (≤3 days) intratumoral presentation of CAR T cells post-treatment is a superior predictor of survival than peripheral persistence. Therefore, we have co-opted IL-8 release from tumors to enhance intratumoral T-cell trafficking through a CAR design for maximal antitumor activity in solid tumors. Here, we demonstrate that IL-8 receptor, CXCR1 or CXCR2, modified CARs markedly enhance migration and persistence of T cells in the tumor, which induce complete tumor regression and long-lasting immunologic memory in pre-clinical models of aggressive tumors such as glioblastoma, ovarian and pancreatic cancer.

[1] Lillian S. Wells Department of Neurosurgery, University of Florida, Gainesville, FL, USA. [2] The Fourth Section of Department of Neurosurgery, the First Affiliated Hospital, Harbin Medical University, Harbin, China. [3] Preston A. Wells, Jr. Center for Brain Tumor Therapy, University of Florida, Gainesville, FL, USA. [4] The Surgery Branch, National Cancer Institute, Bethesda, MD, USA. [5] Department of Pathology, Immunology and Laboratory Medicine, University of Florida, Gainesville, FL, USA. [6] Present address: Legend Biotech USA, Inc., Gainesville, FL, USA. Correspondence and requests for materials should be addressed to J.H. (email: jianping.huang@neurosurgery.ufl.edu)

CAR T-cell therapy is emerging as a promising approach to redirect T cells from cancer patients into tumor-specific killer cells ex vivo[1,2]. This technology has yielded effects in some patients for whom all other treatments have failed, culminating in recent Food and Drug Administration approval for hematologic malignancies[3,4]. However, CAR T therapy in solid tumors is encumbered by numerous challenges, including fewer ideal tumor targets, immunosuppressive tumor-microenvironments (TME), tumor heterogeneity, and minimal migration and persistence of CAR T cells within the tumors.

Finding suitable antigens to target solid tumors is a critical step in successful CAR development. Among the solid tumors, glioblastoma (GBM), ovarian and pancreatic cancer, for example, are often diagnosed at advanced stages, are mostly resistant to existing therapies, and few targeted therapeutics are currently available[5–7]. Thus, identifying clinically precise targets for these and many other cancers is critical. Most of the CAR developments have been involved in a single target, which may result in selective survival and proliferation of antigen-negative tumors due to tumor heterogeneity. Thus, the field is attempting to develop multiple CAR targets[8]. However, in a case study of GBM patient treated with IL-13Rα2 CAR T cells, the authors found that the CAR T cells induce a short-term tumor regression of the GBM expressing ~70% of the IL-13Rα2 antigen[6]. A similar observation was also obtained from our preclinical study using a syngeneic GBM model, in which 70% of the inoculated tumor was positive for CD70, and nearly 40% of the animals were cured by the murine CD70CAR T cells[9]. The results indicate that the endogenous immune system may be activated by the CAR T cell-mediated antitumor responses, which help to trigger an immune response against antigen-negative tumor cells. The treatment outcome of single targeting CARs could be influenced by the target's properties and role in promoting tumorigenesis, immunosuppression, tumor progression, etc. Therefore, the assumption of the effectiveness of a single CAR should be based on each target. The immunosuppressive TME presents challenges for CAR T-cell therapy in solid tumors[10] due to dysfunctional immune surveillance within tumors, such as enriched suppressive elements, metabolic constraints, poor antigen presentation, and chronic T-cell stimulation. Recovering immune surveillance in the TME may provide a significant therapeutic advantage as demonstrated by improved antitumor responses when programmed death 1 receptor (PD-1) antibody is utilized to rescue failed response to CAR T-cell therapy[11], and to improve antitumor response in a preclinical model of solid tumor[12]. It has been shown that the migration and persistence of CAR T cells in peripheral blood are essential factors for durable clinical response in patients with hematologic malignancies[13]. However, peripheral persistence may not be sufficient for the induction of clinical remissions in solid tissue malignancy. Intratumoral CAR T-cell presentation could be a fundamental limitation of a robust response. Since active trafficking of T cells into tumor mass partially depends upon the compatibility between chemokines in tumor and chemokine receptors presented on T cells, mismatched tumor instigated chemokine/chemokine receptor is considered as one of the key factors in limiting T-cell infiltration[14]. A study using adoptively transferred HER-2-specific T cells to treat a metastatic breast cancer patient demonstrated that only 0.1% of the infused T cells were detected in the tumor[15]. Similar results found in preclinical and clinical studies may result from the tumors displaying intrinsic resistance to lymphocyte infiltration and effector function[16,17]. Therefore, intratumoral CAR T-cell infiltration in solid tumors is an area that requires improvement.

Despite all these obstacles that CAR T-cell therapy face in solid tumors, engineered immune responses to cancer have been shown to dramatically improve clinical outcomes in solid tumors[18,19]. Therefore, the goal of this study was to tackle the key obstacles of CAR T-cell therapy in solid tumors. The CD70 CAR platform was employed for improving the clinically relevant tumor models. CD70 is a type II transmembrane protein, and it represents the only ligand for CD27, a glycosylated transmembrane protein of the TNF receptor family[20,21]. Natural CD70 expression is only restricted to extremely activated T and B lymphocytes and a small subset of mature dendritic cells that is rarely observed in cancer patients due to profound immunosuppressive circumstance; however, hematologic malignancies and some solid tumors ectopically express this molecule[9,22,23]. We reported that CD70 tumor expression is an independent indicator of poor survival for patients with lower-grade gliomas (LGG) and GBM, correlating with chemokine-mediated immune inhibition, tumor erosion, proliferation, and migration in GBM[24]. Our studies also indicate that CD70 is involved in macrophage tumor-infiltration and induction of CD8+ T-cell death in GBM[9,24]. These characteristics make CD70 a good target for CAR T therapy in these cancers[9,25,26]. Evidence illustrates that IL-8 enhances recruitment of tumor-associated neutrophils or MDSCs[27], activates epithelial-mesenchymal transition[28], promotes angiogenesis[29,30], and predisposes enhanced resistance, stemness, and metastatic potential[31]. All these pro-tumor properties of IL-8 create significant hurdles for any antitumor treatment modality. In this study, we modified the original CD70CAR with interleukin 8 (also called CXCL8) receptors (CXCR1 and CXCR2, respectively). We utilize tumor-produced IL-8 to guide the IL-8 receptor-modified CD70CAR T cells to migrate into the tumor and induce an enhanced antitumor response in solid tumors.

## Results

**Ionizing radiation enhances IL-8 secretion by tumors**. To determine the effect of ionizing radiation (IR) on a glioma chemokine expression profile, U87 glioma cells were treated with 6 Gy radiation, followed by gene expression detection by a qPCR chemokine array 7 days later. U87 glioma cells express several chemokines (*CCL2*, *CCL20*, *CXCL1*, *CXCL2*, *CXCL8*, etc.), among which IL-8 is the most abundant after radiation (Fig. 1a). To identify the pattern of IL-8 expression by irradiated gliomas, U87 cells were radiated at the doses of 0, 2, and 6 Gy before IL-8 enzyme-linked immunosorbent assay (ELISA) was performed at different time points after radiation. Compared with lower doses, we observed that the highest dose, 6 Gy, induces the most IL-8 secretion at 7 days (Fig. 1b). These results were also reproduced in other glioma cell lines, including primary GBM lines (Fig. 1c), except in pGBM 1156, where lower production of IL-8 protein was found, which may be due to cell death induced by IR. Normal human astrocyte (NHA) did not exhibit similar increases in IL-8 production after IR (Fig. 1d). To determine the impact of radiation on IL-8 gene expression in tumor cells in vivo, an orthotopic murine xenograft model was established by implanting intracranially (i.c.) U87 cells into NSG-B2m mice, followed by fractionated local radiation (Fig. 1e). Enhancement of IL-8 gene expression was observed with increasing doses of radiation (Fig. 1f). Since NRG mice are more tolerant to radiation than NSG-B2m mice, NRG mice were used for subsequent animal studies; thus, the radiation dose was changed to 2 × 4.5 Gy. Immunohistochemistry (IHC) staining was used to confirm the enhancement of IL-8 protein expression in the tumor samples of irradiated mice (Fig. 1g). This radiation-induced IL-8 secretion is not a unique characteristic for gliomas. Multiple tumor types exhibit a similar trend (Supplementary Fig. 1a–f). Therefore, IL-8 production/secretion induced by radiation or expressed naturally can be used as a CAR

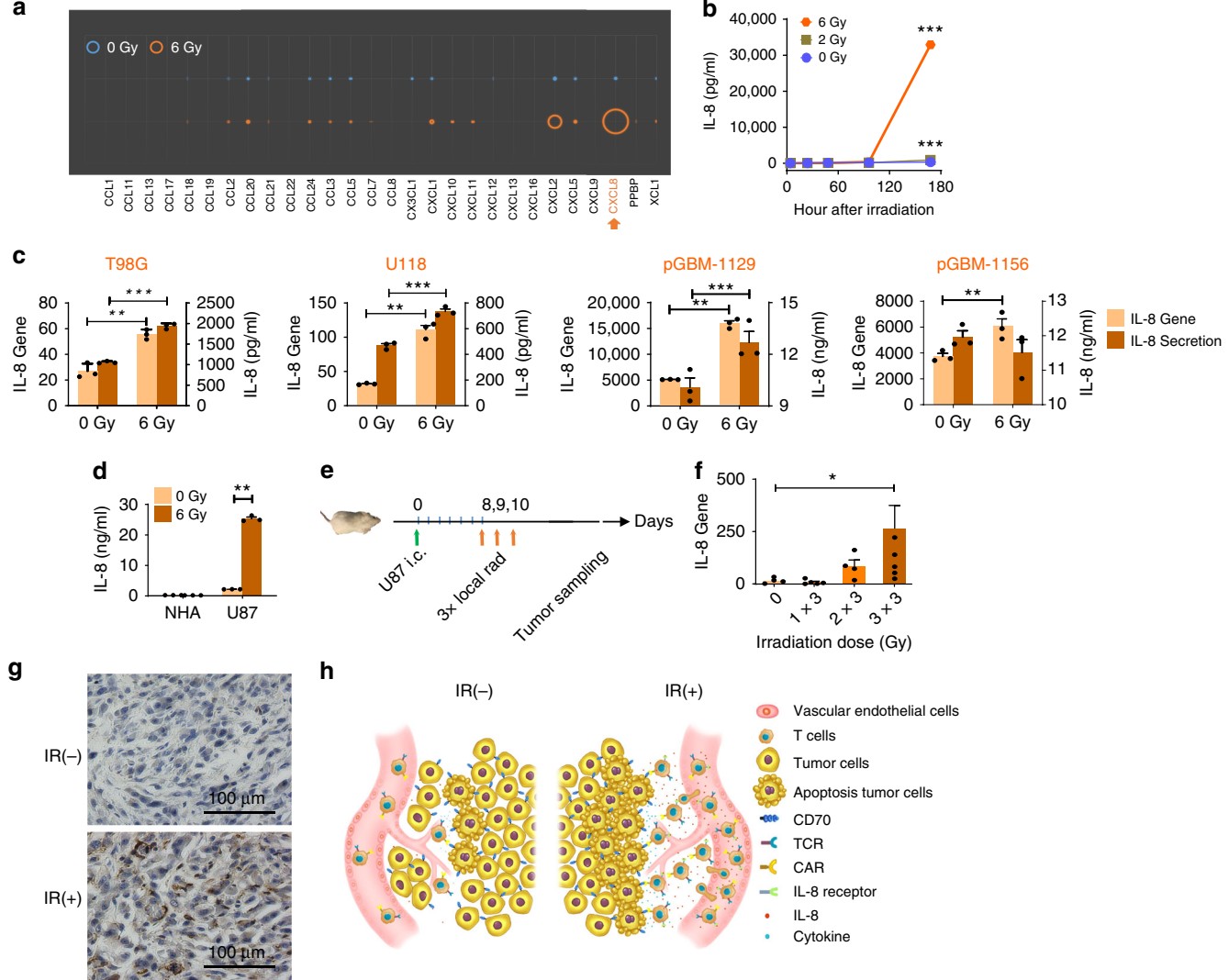

**Fig. 1** IL-8 secretion is enhanced by IR in tumors. **a** Tumor chemokine changes after IR. U87 cells were radiated at 0 or 6 Gy and then seeded at $1 \times 10^4$/ml. Cells were harvested for RNA extraction 7 days later, followed by qPCR for chemokine profiling. The circle size corresponds to the mean values of relative gene expression ($n = 4$). **b** Kinetics of IL-8 secretion by tumor cells. U87 cells were treated with different doses of IR and seeded at $1 \times 10^4$/ml. Supernatants were harvested at 4, 24, 48, 96, or 168 hrs, and the IL-8 secretion was measured by ELISA (mean ± SD; $n = 3$; and the significance was determined by two-way ANOVA). **c** IL-8 production from multiple GBM lines. IL-8 expression was quantified by qPCR and ELISA, respectively, using established glioma lines (T98G and U118) and primary lines (pGBM-1129 and pGBM-1156) 7 days after IR. Tumor cells were irradiated at 0 or 6 Gy and plated at $1 \times 10^4$/ml, and cells and cell supernatants were harvested 7 days later for IL-8 production (means ± SEM; $n = 3$; and the Mann–Whitney $U$ test was used). **d** NHA did not secrete IL-8 after IR. NHA and U87 cells were irradiated, and ELISA was carried out for IL-8 production 7 days after IR (means ± SEM; $n = 3$; and the Mann–Whitney $U$ test was performed). **e–g** IR elevates IL-8 expression in vivo. Design of the xenograft mouse model with local IR (**e**). Mice were injected intracranially with $5 \times 10^4$/mice of U87 cells. Eight days later, mice were locally irradiated with 0–3 fractionated doses daily at 3 Gy/ day. Tumor samples were collected when mice were reaching the endpoint. Tumor IL-8 gene (**f**) and protein (**g**) expression (means ± SEM; $n = 5$; by the Mann–Whitney $U$ test) measured by qPCR or IHC. All the experiments were repeated at least three times. **h** Schema: IR enhances the expression of IL-8 by tumors that can be co-opted by IL-8 receptor-expressing CAR T cells to increase trafficking of CAR T cells to tumors. *$p < 0.05$, **$p < 0.01$, ***$p < 0.001$

T-cell attractor, guiding any IL-8R-modified CAR (8R70CAR) T cells and thus enhancing T-cell tumor migration. In this study, we used CD70CAR, which we have previously shown to induce a potent antitumor function in xenograft and syngeneic GBM mouse models, to test the improvement in antitumor efficacy of the 8R70CAR T cells (Fig. 1h).

**Characteristics of 8R70CAR T cells in vitro.** To exploit radiation-induced IL-8 for enhanced CD70 CAR T chemotaxis, we created an IL-8 receptor (CXCR1 or CXCR2) in tandem with the CD70CARs (8R70CAR: CAR-R1 or CAR-R2). The unmodified

CD70CAR was linked with EGFP (un-mod-70CAR: CAR-EGFP) (Fig. 2a). In addition, vector-transduced T cells were included in the analysis as a control. Neither CXCR1 nor CXCR2 was expressed in quiescent or activated T cells in vitro (Supplementary Fig. 2a–c). The transduction efficiencies of these transgenes were equivalent (Fig. 2b). No differential expansion was detected among the three CARs and vector-transduced T-cell groups during 4 weeks of culture in vitro (Fig. 2c). All groups showed similar CD4 and CD8 populations (Fig. 2d left). Compared to vector-transduced T cells, both modified and unmodified CAR T cells were enriched in an effector memory phenotype (CD45RA⁻ and CD62L⁻) (Fig. 2d middle) and contained a lower expression of exhaustion markers

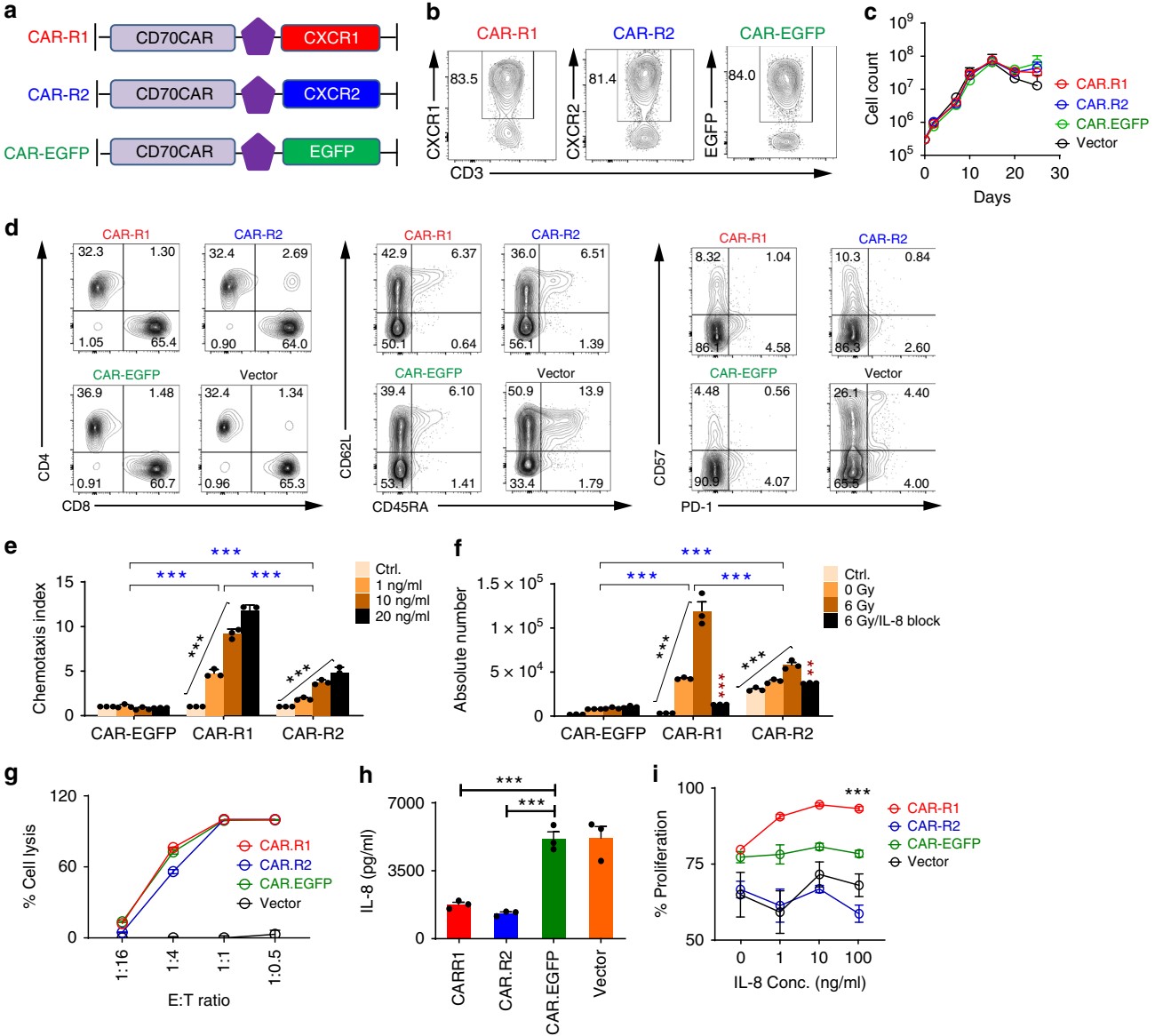

**Fig. 2** Characteristics of 8R70 CAR T cells. **a** Retroviral constructs of 8R CARs and un-mod-70CAR. **b** CAR transduction- efficiency 12 days after transduction. **c** The transduced T cell counts post-transduction in vitro. Transduced T cells were cultured in IL-2-containing medium for 25 days, and the cell counts were measured by a flow cytometer. **d** Representative phenotypical analysis of the CAR T-cells. The cells described in (**c**) were analyzed 15 days after transduction. **e–f** CAR T-cell chemotaxis assay. The U87 cells were radiated at 6 Gy, 7 days later, the supernatant was collected. Anti-IL-8 Ab (2 μg/ml) was added to medium 15 min before the assay, cells from the bottom chamber were counted by flow cytometry 1 hr later, and medium-alone served as control. Chemotaxis index (CI) in (**e**) was defined by the ratio of cells that migrated to the chemoattractants over control. Absolute number of migrated cells in (**f**) was also used to evaluate chemotaxis [means ± SEM; $n = 3$; one-way ANOVA was used among the different amounts of rhIL-8 or doses of IR (black asterisks), two-way ANOVA was used for the different treatment groups (blue asterisks), and Mann–Whitney $U$ test was used for the difference between with and without IL-8 blockade in (**f**) (red asterisks)]. **g** Tumor recognition of the CAR T cells. Luciferase-based killing assay was performed. U87-luciferase cells (U87.Luc, $2 \times 10^4$) were cocultured with CAR T cells as indicated effector-to-target ratio (E:T) (means ± SD; $n = 3$; unpaired $t$ test). **h** CAR T cells decrease IL-8 in vitro. The transduced T cells ($1 \times 10^5$/well) were cultured in T-cell media without IL-2 overnight, and rhIL-8 (5000 ng/ml) was added. The supernatant was collected for IL-8 concentration 24 h later (means ± SEM; $n = 3$; the Mann–Whitney $U$ test). **i** CAR T-cell proliferation in the presence of IL-8. CellTrace-Violet-labeled T cells were stimulated on day 0 with anti-CD3/CD28 Dynabeads (bead: cell = 1:1) in the presence of rhIL-8, the proliferation was measured on day 4 (means ± SD; $n = 3$; the Mann–Whitney $U$ test). ** $p < 0.01$, *** $p < 0.001$

(PD-1 and CD57) (Fig. 2d right), suggesting that the CD70CAR may positively affect the T-cell phenotype. To test IL-8-mediated chemotaxis, Transwell migration-assays were performed using either recombinant human IL-8 or supernatant-derived IL-8 from radiated U87 cell culture. Both CAR-R1 and CAR-R2 T cells showed a rhIL-8 dose-dependent and specific migration (Fig. 2e). The radiated U87 culture media induced migrations of CAR-R1 and CAR-R2 T cells that were abrogated with IL-8 blockade

(Fig. 2f). No differences were observed in cytolytic tumor-killing between the 2 modifications (Fig. 2g). Compared with CAR-EGFP and vector T cells, CAR-R1 and CAR-R2 T cells lowered IL-8 levels in culture assays (Fig. 2h), suggesting the modified T cells may be able to siphon IL-8. Interestingly, IL-8 enhanced the proliferation of CAR-R1 T cells but not CAR-R2 T cells (Fig. 2i). Together, these data suggest that IL-8 can guide the 8R70CAR T cells trafficking into the tumor.

**8R70CARs enhance T-cell trafficking and efficacy.** We next evaluated the migration and antitumor efficacy of the 8R70CARs in vivo using multiple tumor models. First, U87 GBM line was labeled with a near-infrared protein iRFP720[32], to spatially and temporally track tumor growth. Similarly, to track T-cell migration in vivo, the CAR T cells were cotransduced with retroviral vectors expressing click beetle luciferase (CBluc)[33]. NRG mice were implanted i.c. with $5 \times 10^4$ U87.i720, and 10 days later, 2 fractionated local radiation doses were delivered to the tumor site at 4.5 Gy/day. To test the efficacy of the 8R70CARs, a lower dose (1/5) of CAR T cells was used in this study than the highest dose ($1 \times 10^7$) used in our previous report[9]. The CAR-R1, CAR-R2, CAR-EGFP, and vector-transduced T cells were intravenously (i.v.) infused to tumor-bearing mice (17 days after tumor implantation or 7 days after the first radiation) (Fig. 3a). CAR-R1 and CAR-R2T cells migrate more efficiently to the tumor site as early as 2 days after T-cell injection, as measured by CBluc signal, compared with CAR-EGFP. The signal dissipated quickly in the body on day 6 after T-cell infusion in the vector T-cell group (Fig. 3b, d). While the tumor persisted or progressed in the CAR-EGFP and vector-transduced T cell-treated animals, the tumors were nearly undetectable in CAR-R1, and CAR-R2-T-treated groups 6 days after the CAR T-cell transfer (Fig. 3c, e), which translated to long-term survival for the animals (Fig. 3f). To replicate findings in GBM, two additional naturally CD70-expressing tumors, ovarian (SK-OV-3) and pancreatic (PANC-1) cancers, were tested. We found that 100% of the U87 and SK-OV-3 cells, and 58% of PANC-1 cells were CD70 positive (Supplementary Fig. 3a). Tumor recognition was measured in vitro using the CAR-R1, CAR-R2, CAR-EGFP, and vector-transduced T cells by co-culturing with SK-OV-3 or PANC-1 tumor cells. Comparable tumor reactivity was seen among these CAR-transduced T cells against each tumor type according to the results of IFN-γ release (Supplementary Fig. 3b). A similar approach, as described in the GBM model, but different treatment schedules, effector to target (E:T) ratio and follow-ups were performed. In addition, the T cells for the CAR T production were derived from a GBM patient in these models (Fig. 3g, l). We subcutaneously (s.c.) inoculated 20-fold more tumor cells than GBM model but transferred the same amount of CAR T cells into tumor-bearing NRG mice for these two models, therefore applying a lower E:T ratio than in the GBM model. The results indicate a differential tumor trafficking between the 8R70CARs and un-mod-70 CAR T cells as early as 3 days after the therapy. Enhanced luminescence signals were seen in both CAR-R1, and CAR-R2 T cell-treated mice, compared to those detected in the CAR-EGFP or vector T cell-treated mice bearing SK-OV-3 tumors (Fig. 3h upper panels, and 3i) or PANC-1 tumors (Fig. 3m upper panel, and 3n). Moreover, tumor shrinkages were observed in the CAR-R1, or CAR-R2 T cell-treated groups, whereas tumor progressions were seen in the CAR-EGFP and vector-transduced T cell-treated groups (Fig. 3h, lower panels, and 3j), and prolonged survival was achieved in CAR-R1 and CAR-R2 T-cell groups in the SK-OV-3 model (Fig. 3k). Similar results were attained in the PANC-1 model (Fig. 3l–p). In summary, these data indicate that our CAR modifications elicit significantly enhanced CAR T cell tumor trafficking and antitumor effects.

**More activated 8R70CAR T cells in the tumor.** Next, we wanted to confirm that the IVIS signals in the ROI were illustrative of the intratumoral CAR T cells (Fig. 4a). The U87 GBM model was used with T cells derived from a GBM patient for CAR T-cell production. Due to the short interval between the start of treatment and tumor regression, we decided to evaluate the intratumoral CAR T-cell phenotype at 2 days post-treatment. The results illustrated that the

intratumoral CAR T cells measured by CD45[+] cells matched the IVIS luminescence signals. Substantially more tumor-infiltrating CAR T cells were detected in CAR-R1, and CAR-R2 than CAR-EGFP or vector-transduced T cells (Fig. 4b). The majority of these intratumoral CAR T cells express granzyme B (GZMB), and the tumors treated with CAR-R2 T cells showed a slightly higher % of GZMB[+] T cells. Although there was a trend that the CAR-EGFP group showed relatively higher PD-1 level than the CAR-R1 and CAR-R2 on average, no significant statistical difference was noted among groups (Fig. 4c–e). Furthermore, we found a good correlation between the luminescence signals and CD45[+] CAR T-cell density (Fig. 4f). In summary, our CAR modification significantly enhances intratumoral T-cell migration, and these cells present the activation phenotype.

**Exhausted/anergic un-mod-70CAR T cells in the tumor.** We observed that while animals receiving CAR-R1 and CAR-R2 T cells remained tumor-free, tumor relapses were seen in the CAR-EGFP T cell-treated group. All three groups showed CAR T-cell signals in tumor and peripheral circulation (from 22 days after T-cell transfer) (Fig. 5a), however, tumor relapse was first observed from day 36 in the CAR-EGFP group (Fig. 5b). To comprehend the basis of relapse, tumors, and spleens were harvested for T-cell phenotypical analysis when relapsed animals reached the endpoint (Fig. 5c). Tumor-infiltrating T cells (TIT) presented a similar CD8/CD4 ratio, compared with the preinfusion T cells (baseline, BL), but the ratio from the spleen (SP) was relatively lower in these mice (Fig. 5d). The phenotypical analysis suggests that a majority of the T cells in these mice bear a CD45RA[−] CD62L[−] effector memory phenotype (Fig. 5e). A greater decrease in CAR T cells (~20%) in the tumor than in the spleen was also revealed (Fig. 5f). In addition, compared with BL, the TIT and SP expressed higher PD-1 (~90% positivity) (Fig. 5g). We then restimulated these T cells with U87 tumors in vitro, but no reactivity was detected by IFN-γ release (Fig. 5h). No significant change was found on the relapsed tumors for CD70 expression (Fig. 5i), but we did appreciate a slight increase (no statistical difference) in PD-L1 on the relapsed tumors (Fig. 5j). To determine whether the T cells could rescue the animals, 2 relapsed mice were treated again with $2 \times 10^6$ of CAR-EGFP T cells without local radiation. Although CAR T cells gradually accumulated in the ROI (Fig. 5k, l), the tumor eventually progressed, and all mice died of their respective tumors (Fig. 5m). Taken together, these results suggest that inadequate intratumoral CAR T-cell trafficking allows tumors to take control of the microenvironment that dysfunctions the T cells.

**Late and long-term effects of 8R70CAR T-cell therapy.** To assess the late and long-term effects of the 8R70CAR T cells, animals with large/late-stage tumors were tested. The same GBM tumor model as that shown in Fig. 3a was used, but all treatment time points were sequentially delayed for 12 days (e.g., the CAR T cells were given 29 days after the tumor implantation), with the median survival time of the previous mouse model being 35 days. Compared to animals receiving CAR-EGFP T cells, CAR-R1 and CAR-R2 T cell treated groups show even more significantly enhanced trafficking, tumor regression (Fig. 6a–d) and survival (Fig. 6e). Stronger signals of T cells in the periphery were seen in the CAR-EGFP group (Fig. 6f), which did not associate with tumor regression. The results prompted us to determine further if the imaging signals are predictive of survival. Interestingly, CAR T-cell signals measured by IVIS in the ROI rather than in the periphery as early as 1 day post-treatment were found to be associated with survival (when data of 1–6 days post-treatment were evaluated) (Fig. 6g). This result was also confirmed by the ovarian and pancreatic cancer models using the data from Fig. 3

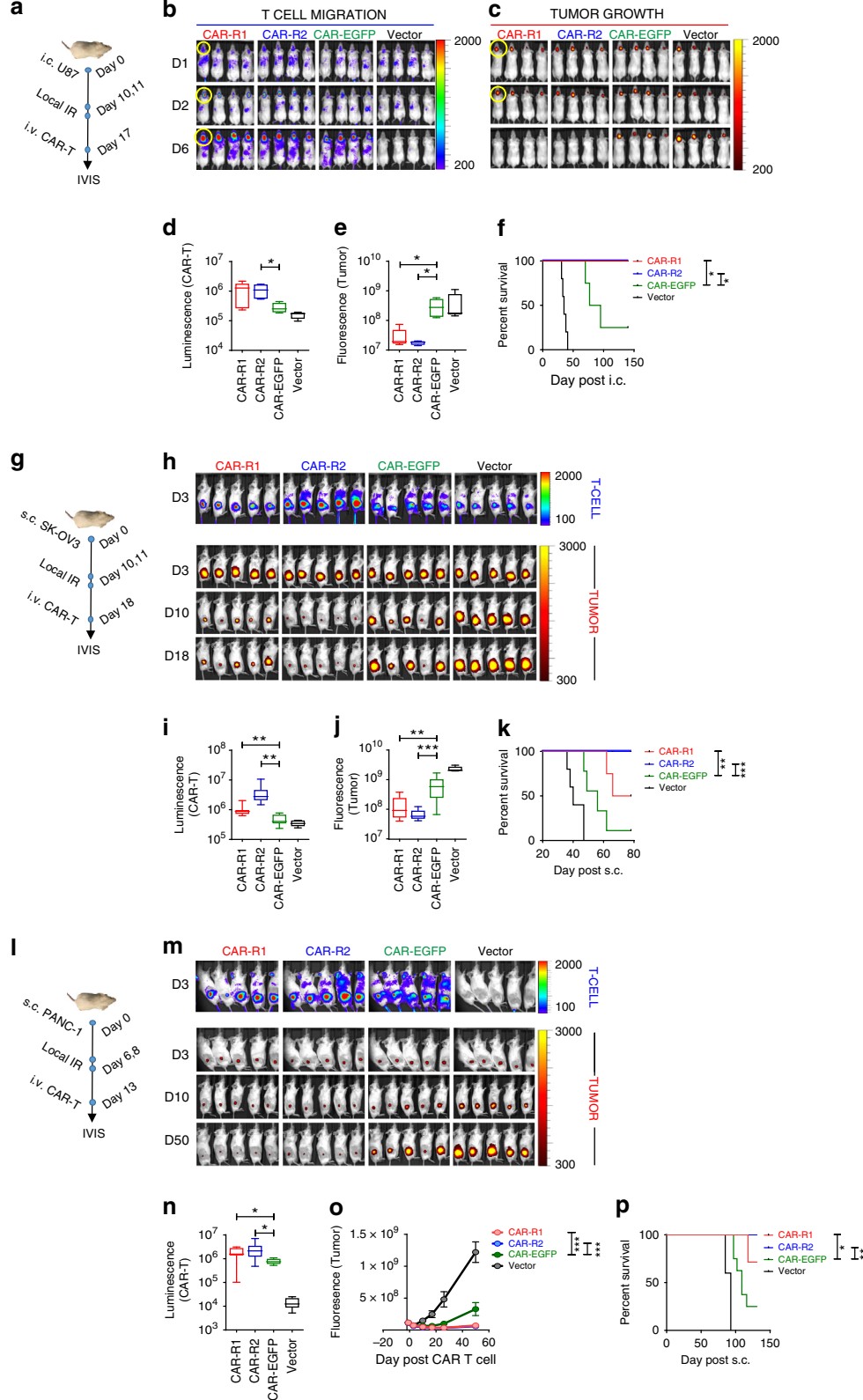

(Fig. 6h). To further evaluate the long-term protection by 8R70CAR T cells, the cured mice were rechallenged in the opposite hemisphere with U87.i720 (on day 77 after the first tumor i.c. inoculation) (Fig. 6i, j). While the rechallenged mice grew tumors that regressed completely in the CAR-R1 or CAR-R2-treated groups, all the tumor-only mice succumbed to their disease characterized by progressing tumors (Fig. 6k). A remarkable surge of the CAR-R1 or CAR-R2 T cells was observed after tumor injection in rechallenged sites (Fig. 6l), which precipitated long-term survival benefits (Fig. 6m). Thus, the new CAR design equips T cells with better antitumor reactivity and immunologic memory for long-lasting tumor control.

**Fig. 3** 8R70CARs enhance T-cell trafficking and antitumor efficacy. **a** The treatment plan in the GBM model. i720 transduced U87 tumor cells ($5 \times 10^4$/mouse) were implanted i.c. on day 0, followed by fractionated local IR on day 10 and 11 (4.5 Gy/day) and i.v. administration of the CARs or vector T cells on day 17. IVIS images for luminescence (T cells) and fluorescence (tumors) were taken. **b, c** T-cell migration and tumor growth at a region of interest (ROI, highlighted with yellow circles) 1, 2, and 6 days after the CAR T-cell treatment. **d, e** Quantification of T-cell migration to the ROI, 2 days after the T-cell transfer from (**b**) and tumor growth, 6 days after the T-cell transfer from (**c**). **f** Survivals ($n = 4$–5/group) from (**b**) and (**c**). **g** The treatment plan in an ovarian cancer model. i720 transduced SK-OV-3 tumor cells ($1 \times 10^6$/mouse) were implanted s.c. on day 0, followed by fractionated local IR on day 10 and 11 (4.5 Gy/day) and the T cells i.v. on day 18. **h** CAR T cells (upper panel) and tumor size (lower panels) in the ROI, after the CAR T-cell transfer. **i** CAR T-cell migration to ROI 3 days after the transfer (**h**, upper panel). **j** Tumor size 10 days after the transfer (from **h**, lower panel). **k** Survivals ($n = 5$–9/group) from (**h**). **l** The treatment plan in a pancreatic cancer model. i720 transduced PANC-1 tumor cells ($1 \times 10^6$/mouse) were implanted s.c. on day 0, followed by fractionated local IR on day 6 and 8 (4.5 Gy/day) and i.v. injection of the T cells on day 13. **m** Imaging of the CAR T cells in ROI (upper panel, 3 days post T-cell transfer) and tumor size (lower panels) after the CAR T-cell transfer. **n** CAR T-cell migration in ROI, 3 days after the transfer. **o, p** Tumor volumes and survivals after the CAR treatment. The boxplots represent the mean and minimum-to-maximum range. The Mann–Whitney $U$ test for (**d, e, i, j, n**), two-way ANOVA for (**o**), and the log-rank test for (**f, k, p**). *$p < 0.05$, **$p < 0.01$, ***$p < 0.001$

## Discussion

Despite promising preclinical investigations[9,26,34–36], CAR T-cell therapy remains encumbered by significant hurdles in advanced solid cancers[6,7]. Previously, our group has tested the CD70CAR in clinically relevant human xenograft and syngeneic GBM models. We demonstrated a dose-dependent antitumor response using the CD70CAR T cells. Complete tumor regression was only observed in the highest CAR T-cell dose ($1 \times 10^7$)[9], which is equivalent to the maximal dose that can be practically used in cancer patients. Therefore, in this study, we attempted to improve the efficacy of the CD70CAR. One of the barriers of CAR T-cell therapy is inadequate CAR T-cell trafficking to tumor foci, which is a complex interaction between T cells and endothelial cells[37,38], and mismatched chemokines from tumor cells and cognate receptors on T cells are responsible in part for the insufficient homing of CAR T cells to tumors[38]. We sought to enhance this technology by integrating chemokine receptors into a novel CAR design for improved intratumoral trafficking.

The results from in vitro experiments suggest that IL-8 is a chemokine that can be leveraged for CAR T-cell therapy. Tumors can overexpress IL-8 naturally, and ionizing radiation was found to significantly increase the production of IL-8 in tumors[39]. Our data show that although IL-8Rs modification did not alter the phenotype and tumor recognition of the transduced T cells, it was capable of enhancing T-cell chemotaxis in vitro when IL-8 was present. The 8R70CARs also showed superior properties compared to the un-mod-70CAR T cells in animal models. The modified CAR T cells had improved intratumoral migration and more robust antitumor responses in GBM, ovarian, and pancreatic cancer models. The enhanced trafficking exhibited by these 8R70CAR T cells is a requisite for regressions of these tumors. Two days post-CAR T treatment, the intratumoral 8R70CAR T cells showed activated phenotypes, such as high GZMB and moderate PD-1 expression that may be caused by activation. However, delayed trafficking or lower intratumoral presentation of T cells with the un-mod-70CAR promotes the formation of a suppressive intratumoral micro-environment with dysfunctional CAR T cells, resulting in tumor relapse despite the persistence of the CAR T cells in the periphery and the maintenance of CD70 expression on tumor cells. More CD62L⁻ CD45RA⁻ CAR T cells bearing an exhausted (i.e., high CD57 and PD-1 expression) phenotype and anergic status upon recurrence, and decreased intratumoral CAR T-cell presentation were observed in the relapsed tumors. The additional administration of the un-mod-70CAR T cells did not save the animals, possibly because the minimal intratumoral CAR T cells convey help that is too little and too late. These data highlight the deleterious effects of the solid tumor microenvironment and limited CAR T-cell trafficking on CAR T-cell therapy.

The ability to treat late-stage or large tumors is potentially life-saving for patients for whom surgical removal is not an option. The 8R70CAR T cells exhibited enhanced migratory and pro-liferative capacities leading to complete tumor regression of larger and late-stage tumors than the un-mod-70CAR T cells. Five days after tumor rechallenge, a surge in CAR T cells in the tumor site was detected, and long-term protection was obtained. In addition to improving the efficacy of the CAR T technology, several clinical implications and further investigations of this study need to be highlighted. Firstly, previous reports have provided evidence that peripheral T-cell persistence correlates with clinical responses in adoptive T-cell transfer therapy using tumor-infiltrating lymphocytes and CAR T-cell therapy for hematopoietic malignancies[3,40–45]. However, the results of this study demonstrated a different perspective. Although more un-mod-70CAR T cells were observed in the periphery 4–6 days after the T-cell transfer than 8R70CAR T cells, this did not translate to an enhanced antitumor response. Intriguingly, our results demonstrate that as early as 1 or 3 days after the therapy, the levels of intratumoral persistence of the CAR T-cell were found to correlate with overall survival. This observation is thought-provoking: in the future, we may be able to predict treatment outcomes shortly after CAR T-cell transfer using noninvasive imaging such as PET/CT or other innovative imaging methods to detect intratumoral CAR T cells[46,47]. Secondly, co-opting IL-8 to enhance CAR T-cell trafficking using local radiation or other modalities that could increase IL-8 secretion may be harnessed for different tumor types[48]. Radiation also elevates CD70 expression on the tumor, potentially explaining why more CD70-positive tumors were discovered in recurrent tumors than primary tumors[9]. Thus, standard therapies inadvertently help make CD70 a constant target for both disease states, which we can and should take advantage of. Finally, IL-8 is considered to be a pro-cancer chemokine with a role in tumor immunosuppression, and we found that the 8R70CAR T cells appear to be able to siphon IL-8 in vitro. We hypothesize that these CAR T cells may act as a sink to neutralize or remove IL-8 from the intratumoral micro-environment after trafficking into the tumor, and provide an advantage for improved antitumor therapeutics by lowering the intratumoral IL-8, in addition to enhancing T-cell trafficking. Interestingly, we also found that CAR-R2 is superior to CAR-R1 in vivo, although CAR-R1 displayed better chemotaxis and increased proliferation in response to IL-8 in vitro. These conflicting observations remained consistent in ovarian and pancreatic cancer models as well. The mechanism regarding unique immunological functions between these two modifications is unclear and requests more experimentation. A limitation of this study was the lack of a testing approach in a syngeneic model due

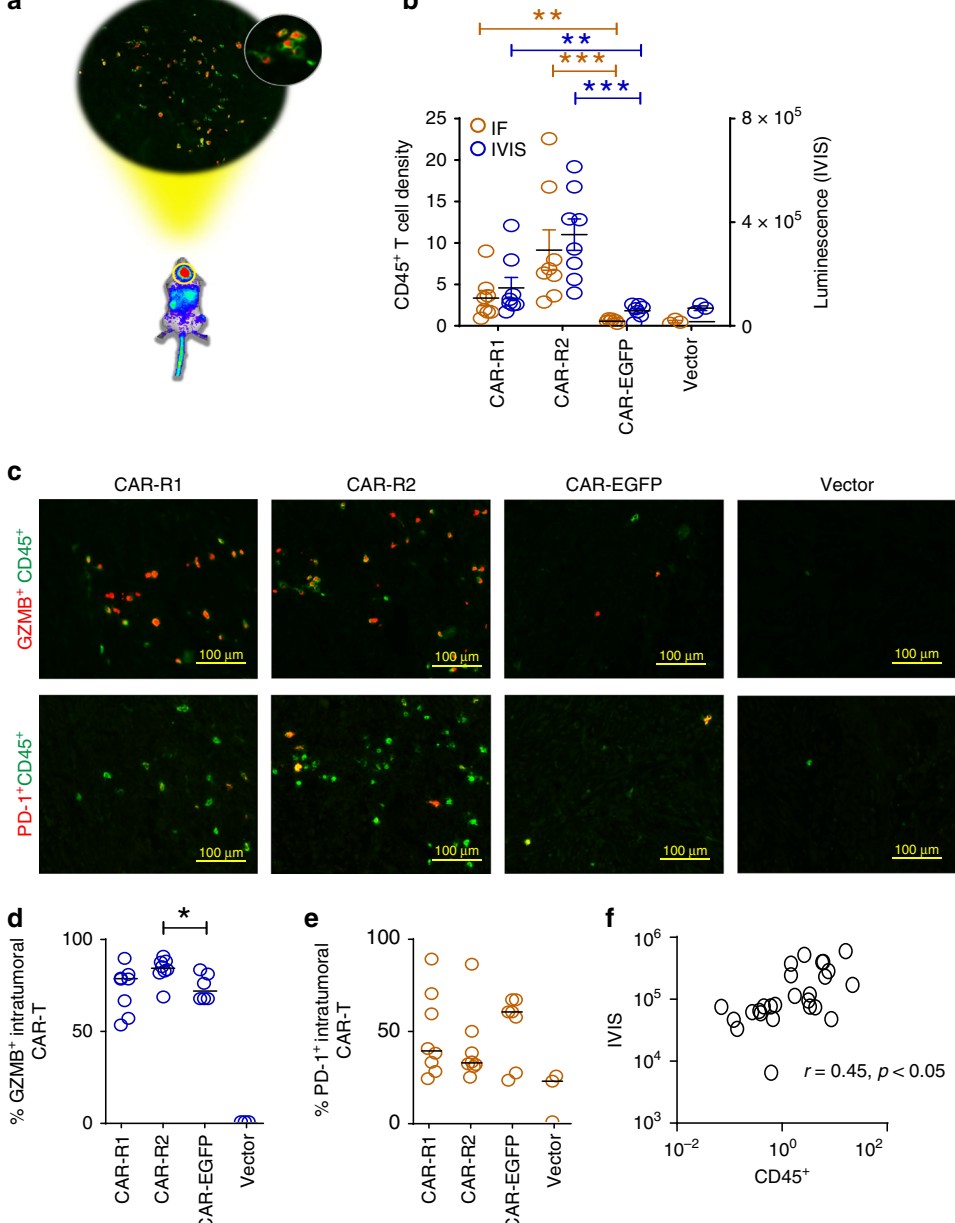

**Fig. 4** Increased intratumoral CAR T cells with activation phenotype in the 8R70 CAR T-treated tumors. **a** A schema of the intratumoral T-cell phenotype. A similar approach as described in Fig. 3a was used. The T cells for the CAR transduction were derived from a GBM patient. Two days after the CAR T-cell transfer, IVIS imaging was carried out for confirming the existence of CAR T cells in the tumor area, and then the tumors were dissected from the animals, followed by immunofluorescence staining for phenotype analysis of tumor-infiltrating CAR T cells. **b** Intratumoral CAR T-cell infiltration in these treatment groups. The readouts of intratumoral CAR T cells measured by CD45[+] CAR T-cell density (numbers/mm$^2$) and luminescence signals in ROI quantified by IVIS from the same animal are shown (mean ± SEM, the Mann–Whitney $U$ test). **c** The phenotype of the intratumoral CAR T cells. The tumor sections were stained with two antibodies against GZMB (red, upper panel) and CD45 (CAR T cells, green), or PD-1 (red, lower panel) and CD45 (green). Representative images are displayed for the treatment groups. **d**, **e** Quantification of the frequency of GZMB[+] and PD-1[+] intratumoral CAR T cells. The cell density of CD45[+], GZMB[+], and PD-1[+] cells (numbers/mm$^2$) in the IF images were acquired from the entire tumor mass. The percentage of CD45[+]GZMB[+] and CD45[+]PD-1[+] CAR T cells were, respectively, calculated based on the cell density of GZMB or PD-1 over CD45. The bars indicate the median values. **f** A correlation between IF and luminescence signals. The correlation was determined by Pearson correlation. *$p < 0.05$, **$p < 0.01$, ***$p < 0.001$

to the fact that the gene encoding IL-8 is absent in mice[49,50]. However, the treatment of multiple human tumor xenografts using healthy donor- and GBM patient-derived CAR T cells provides proof of concept for future human studies. Thus, our results demonstrate the feasibility and efficacy of utilizing the 8R70CAR T cells for homing to the tumor site.

In summary, we used treatment-resistant tumor models to demonstrate that utilizing naturally expressed or radiation-induced IL-

8 release from the tumor can enhance intratumoral T-cell trafficking. Rapid CAR T-cell intratumoral migration may efficiently eliminate tumor growth and reverse tumor-induced immunosuppression, leading to maximal antitumor response. In addition, our results hold potential relevance for the treatment of other CD70[+] malignancies and for understanding the role of CD70 in tumor progression. Currently, testing CAR modifications in syngeneic systems is ongoing, and a human phase I clinical trial in GBM is on the way.

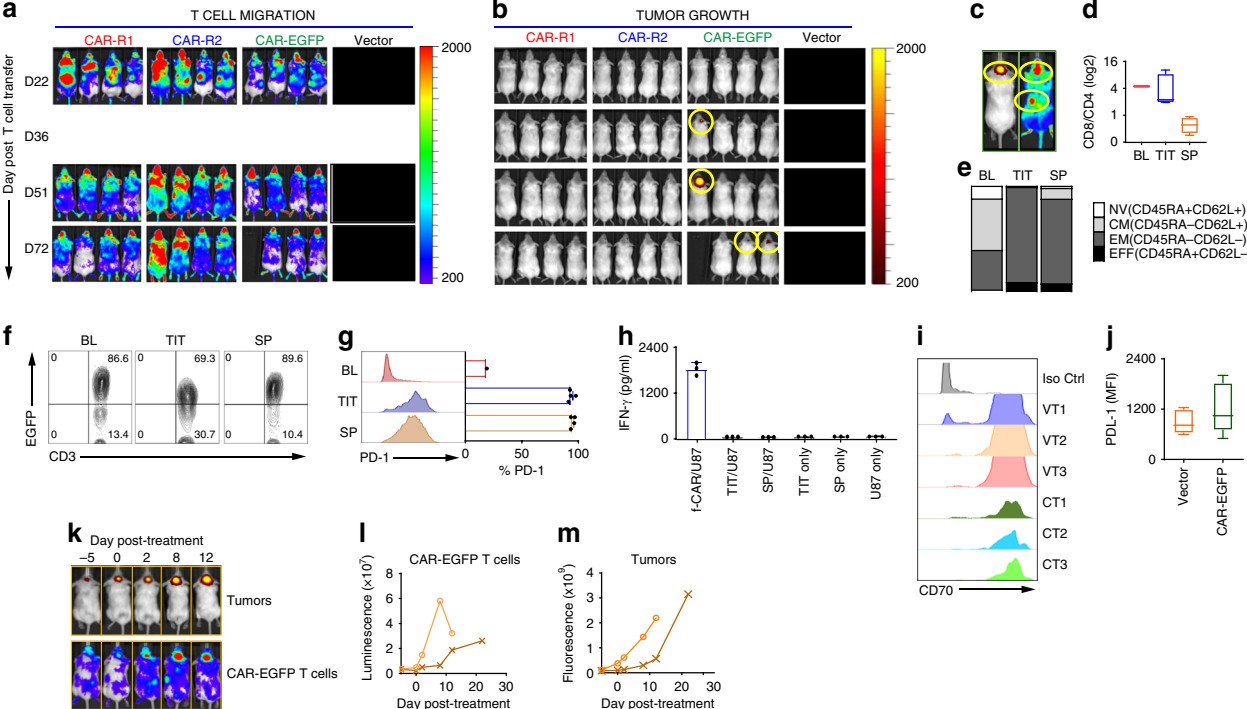

**Fig. 5** un-mod-70CAR T-cell exhaustion at the time of tumor relapse. **a, b** T-cell persistence and tumor progression after the CAR T-cell transfer. Images of CAR T-cell and tumor growth were taken after cell transfer. Tumor signals were undetectable on day 22 in all CAR T cell-treated groups shown in Fig. 3; tumor relapses were observed from day 36 only in CAR-EGFP T cell-treated mice. **c** Representative images of the tumor (left) and CAR T cells (right) at the endpoint of a relapsed mouse. Tumors and spleens from these mice were harvested and analyzed by flow cytometry. **d, e** CD8/CD4 ratio, [Naïve (NV), Central Memory (CM), Effector Memory (EM), Effector (EFF)] of the tumor-infiltrating T cells (TIT, $n = 4$) and spleen (SP, $n = 4$), compared to before adoptive transfer (baseline, BL). The human $CD3^+$ T cells were gated. **f** CAR T-cell persistence post CAR T cell transfer. The CAR T cells in BL, TIT and SP were determined. The data are representative of three mice. **g** PD-1 expression on $CD3^+$ T cells in BL, TIT, and SP (mean ± SD, $n = 4$). **h** Tumor recognition of the TIT and SP cells isolated from the relapsed mice when they reached the endpoint. Fresh U87 tumors ($1 \times 10^5$) were used as the target, and fresh-made CAR-EGFP (f-CAR) T cells were used as a positive control. IFN-γ release assay was carried out 18 hrs after the coculture (means ± SD). **i, j** Expressions of CD70 and PD-L1 on relapsed tumors. The relapsed mice ($n = 3$) from the CAR-EGFP-treated (CT) and vector T cell-treated (VT) group were evaluated for CD70 and PD-L1 expressions by flow cytometric analysis. **k** Representative images of tumor progression and CAR T-cell persistence in the ROI are displayed. **i–m** The relapsed mice ($n = 2$) were re-administered the fresh-made CAR-EGFP T cells by i.v. ($2 \times 10^6$/mouse) without IR, and the CAR T cells and tumor growth were monitored and evaluated in these mice. The boxplots represent the mean and minimum-to-maximum range

## Methods

**Cell lines and cell culture.** Healthy donor blood samples were obtained from LifeSouth Community Blood Centers (Gainesville, FL). The GBM blood samples were derived from non-adherent cells of PBMC that were no longer needed for GMP manufacture and would otherwise be discarded. The human glioma cell lines U87 and human pancreatic carcinoma cell line PANC-1 were purchased from ATCC (Manassas, VA). Normal human astrocytes were purchased from Lonza (Basel, Switzerland). Human glioma cell lines T98G and U118, human ovarian cancer cell line SK-OV-3, human prostate cancer cell line PC-3, human breast cancer cell line MDA-MB-231, human lung carcinoma cell line A549, and human T-cell leukemia cell line Jurkat were gifts from multiple institutions of the University of Florida (Gainesville, FL). GP2-293 for retroviral packaging was from Clontech Laboratories (Mountain View, CA), and 293T/17 for lentiviral packaging was from ATCC. The human primary glioblastoma-derived cell lines pGBM-1129 and pGBM-1156 were obtained from The Florida Center for Brain Tumor Research (FCBTR). For imaging purposes, the U87, PANC-1, and SK-OV-3 were respectively transduced by lentiviral vectors to overexpress iRFP720. GP2-293 and 293T/17 cells were cultured in DMEM (Thermo Fisher Scientific) containing sodium pyruvate with 10% fetal bovine serum (FBS, VWR) and 1% Penicillin–Streptomycin (Thermo Fisher Scientific). Jurkat cells were cultured in RPMI 1640 medium (Thermo Fisher Scientific) with 10% FBS and 1% Penicillin–Streptomycin. All other established tumor cell lines were cultured in DMEM with 10% FBS and 1% Penicillin–Streptomycin. NHA cells were cultured in AGM$^{TM}$ Astrocyte Growth Medium (Lonza). Human primary glioblastoma-derived cell lines were maintained in a complete neural stem cell (NSC) medium, which is a mixture of a NSC basal medium and NSC proliferation supplement at a 9:1 ratio (STEMCELL Technologies Inc.) with 20 ng/ml EGF (R&D Systems), 10 ng/ml bFGF (STEMCELL Technologies Inc.), 1 µl/ml of 0.2% heparin (Sigma-Aldrich), and 1% Penicillin–Streptomycin.

**Radiation in vitro and in vivo.** For in vitro experiments, tumor cells were trypsinized to single-cell suspensions and then radiated by X-RAD 320 (Precision X-Ray), a biological X-ray irradiator, at 1 Gy/min at room temperature. Totally, $2 \times 10^5$ radiated cells, and control cells were seeded in T75 flasks for subsequent assays.

Intracranial tumor local radiation was performed with a Varian Clinac 600 C system (Varian Medical Systems). Tumor-bearing mice were anesthetized with 100 mg/kg ketamine (Sigma-Aldrich) and 10 mg/kg xylazine (Sigma-Aldrich) by intraperitoneal injection and placed 100 cm away from the radiation source as the standard treatment with the radiation beam being sharply collimated to only expose the skulls to the radiation beam. A bolus (a blanket of 1 cm thickness tissue-equivalent material) was placed over the mice and in contact with the head. The bolus is needed to allow the radiation to build to the proper dose below the surface of the skull (i.e., within the intracranial tumor). With the bolus, the radiation dose will be uniform to within 5% of the mean head dose. A dose of 3 or 4.5 Gy to the right side of the head was delivered. Fractionated radiation was performed as indicated. Subcutaneous tumor local radiation was performed with X-RAD 320 (Precision X-ray). The tumor-bearing mice were anesthetized in the same way as described in the intracranial model. Mice were placed side-by-side in the prone along the borders of the square cone, in a position that only allowed the animal's right-hind flank to be irradiated, while the rest of body was outside of the radiation field. A dose of 4.5 Gy was delivered, and fractionated radiation was performed.

**Quantitative real-time PCR.** Total RNA was isolated using the RNeasy Mini Kit (Qiagen) and reverse transcribed using the iScript Reverse Transcription Supermix (Bio-Rad Laboratories). A predesigned 384-well panel for human chemokines (Bio-Rad Laboratories) with SsoAdvanced Universal SYBR Green Supermix (Bio-Rad Laboratories) was used to determine the chemokine profile of the radiated glioma cell line. TaqMan Gene Expression Assays (ThermoFisher Scientific) were used

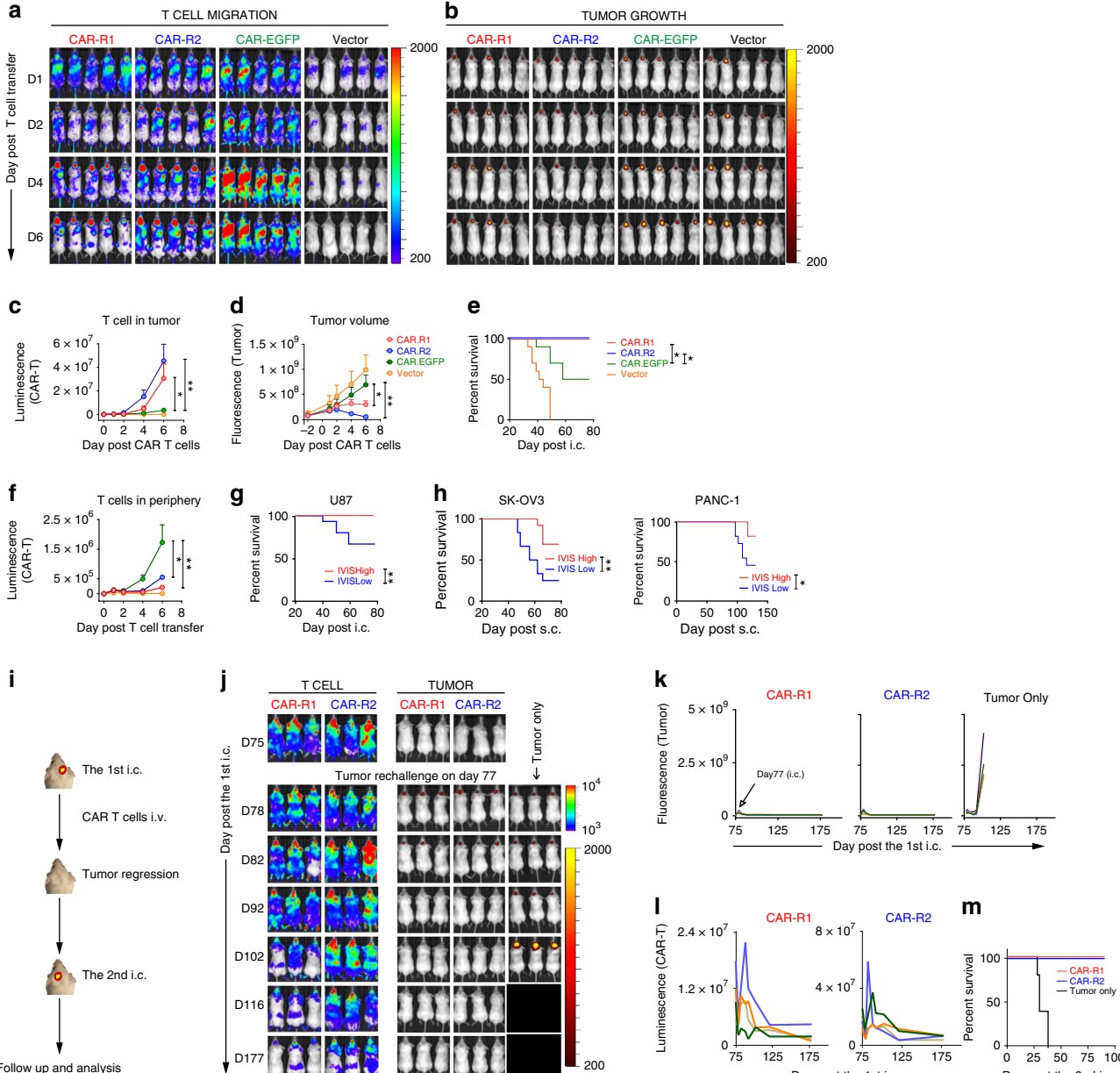

**Fig. 6** 8R70CAR T cells cure late-stage tumors and provide long-term protection. **a**, **b** The 8R70CARs and un-mod-70CAR T cells in treating a late-stage tumor model. Compared with the experiment in Fig. 3a, the local radiation was delayed until day 22 and day 23, followed by CAR T-cell injection i.v. on day 29. Images show CAR T-cell tumor trafficking and tumor growth after adoptive transfer. **c**, **d** Kinetics of T-cell tumor trafficking and tumor growth at ROI. Data represent means ± SEM. **e** Survival of CAR-R1, CAR-R2, CAR-EGFP, and vector-treated mice. Kaplan–Meier survival curves are displayed. **f** Quantitation of T-cell signals in the periphery using the data shown in (**a**). **g** The correlation between early IVIS imaging signals post-treatment and survival in GBM. Mice of all three CAR T cell-treated groups described in (**a**) were stratified into low and high groups based on the median value of luminescence signals by IVIS, 1 day post-treatment. The survival difference between the groups was determined by Kaplan–Meier survival analysis. **h** Confirmative analysis of the correlation between early T-cell signals (3 days post-treatment) and survival using the results of the ovarian and pancreatic tumor models described in Fig. 3i, k, n, and p. **i** Experimental schema of tumor rechallenge. **j** Representative images of cured mice and tumor rechallenge. The cured mice treated by the CAR-R1, CAR-R2 T cells were i.c. inoculated with fresh tumors on the contralateral site with $5 \times 10^4$/mouse U87.i720 on day 77 after first tumor implantation. Tumor-only controls represent naïve mice with the same tumor implantation. **k**, **l** Tumor growth and CAR T-cell tumor trafficking following tumor rechallenge. Each line represents a single mouse. **m** Kaplan–Meier survival curves of the rechallenged mice (four mice/group). The Mann–Whitney $U$ test was used, and the log-rank test was used for the survival analysis. *$p < 0.05$, **$p < 0.01$

specifically for IL-8 (Hs00174103_m1) gene expression with SsoAdvanced Universal Probes Supermix (Bio-Rad Laboratories) according to the manufacturer's protocols, and RPL13A (Hs01578912_m1) was used for normalization. qPCR runs were performed on the QuantStudio™ 12 K Flex Real-Time PCR System (ThermoFisher Scientific) or CFX96 Touch™ Real-Time PCR Detection System (Bio-Rad Laboratories). Relative quantification of gene expression was carried out via the delta Ct methodology.

**Immunohistochemistry**. Mouse brain tumor samples were immediately fixed in 4% PFA or 0.2% formalin, and cryoprotected by 15% and then 30% sucrose solutions. All samples were embedded in OCT, slowly frozen in a slush of dry ice and 100% alcohol, and then cryosectioned (5 or 10 μm). Intratumoral IL-8 expression test was carried out using IHC, performed by the Molecular Pathology Core at the University of Florida. Briefly, the Dual Endogenous Enzyme (Agilent) was used to quench endogenous peroxidase activity. To reduce nonspecific background staining, 2.5%

normal goat serum was used. Sections were incubated with rabbit anti-human IL-8 (Abcam) for 1 h at room temperature. The stain was visualized using goat anti-rabbit HRP polymer (Vector Laboratories Inc.), DAB chromogen (Biocare Medical), and CAT hematoxylin counterstain (Biocare Medical). Immunofluorescence (IF) was performed for intratumoral CAR T-cell phenotyping. Whole tumor sections were stained with primary antibodies: rat anti-human CD45 (Thermo Fisher Scientific), rabbit anti-human Granzyme B (Abcam), or rabbit anti-human PD1 (Novus Bio-logicals) at 4 °C for overnight, and followed by secondary antibodies coupled with a fluorophore or HRP (goat anti-rat IgG-Alexa Fluor488 and goat anti-rabbit IgG-HRP, ThermoFisher Scientific) at room temperature for 2 h. HRP activity was revealed using Tyramide-Alexa Fluor568 (ThermoFisher Scientific), and DAPI (Thermo Fisher Scientific) was used for nuclear staining. All IHC or IF antibodies were listed in Supplementary Table 2. For imaging quantitation, the entire tumor was scanned by BZ-X810 All-in-One fluorescence microscope and analyzed with BZ-X800 Analyzer software (Keyence). Results represent a cell density, which is calculated as an absolute number of positive cells per square millimeter.

**IL-8 production**. Tumor cells were exposed to X-rays at the dose indicated. Culture supernatants were harvested after 7 days, and ELISA was performed to measure the concentrations of IL-8 using the IL-8 ELISA kit (eBioscience) according to the manufacturer's protocols.

**Analysis of cytokine production of effector cells**. To assess the tumor recognition of CAR T cells, T cells were cocultured with tumor cell lines, $1 \times 10^5$ effector cells or control T cells were cocultured with $1 \times 10^5$ target tumor cells in each well of the 96-well plate. Cell culture supernatants were harvested and assayed 18 h later for IFN-γ detection by ELISA (R&D system). The culture supernatants were diluted to be in the linear range of the assay.

**Retroviral and lentiviral constructs**. pMSGV8 plasmid served as the backbone of all modifications. Human IL-8 receptors CXCR1 and CXCR2, and EGFP cDNAs were synthesized by Integrated DNA Technology (IDT, California, USA) and then respectively subcloned into downstream of the CD70CAR, which was linked with a modified 2A peptide to generate constructs of pMSGV8-CD70CAR-CXCR1, pMSGV8-CD70CAR-CXCR2, and pMSGV8-CD70CAR-EGFP (designated as CAR-R1, CAR-R2, and CAR-EGFP). A furin cleavage site followed by a V5 peptide tag was introduced to the P2A peptide to improve gene expression[51]. CBluc was synthesized by IDT and subcloned into pMSGV8 to generate pMSGV8-CBluc. iRFP720 was synthesized by IDT and subcloned into pD2109-EFs (ATUM) plasmid, a lentiviral vector, to generate pD2109-EFs-iRFP720. pLenti-CMV-Puro-LUC (Addgene) was used to generate lentiviral particles expressing firefly luciferase.

**Transduction of human T cells and tumor cell lines**. VSV-G-pseudotyped viral particles encoding CD70 CAR with different modifications were produced by transient transfection of GP2-293 cells. To produce retrovirus, GP2-293 cells were transfected with 2 μg pMD2.G (Addgene) and 2 μg transfer plasmid using 10 μl Lipofectamine 2000 (Thermo Fisher Scientific) in 600 μl OPTI-MEM (Thermo Fisher Scientific). Two days later, the supernatants were harvested for CAR transduction. Human peripheral blood mononuclear cells (PBMCs) from healthy donors or GBM patients were obtained. T cells from PBMCs were activated with anti-CD3/anti-CD28 Dynabeads (Thermo Fisher Scientific) (cell-to-bead ratio = 1:3) 72 h before the transduction. For some experiments, T cells were cotransduced with additional viral particles expressing CBluc. RetroNectin (Clontech Laboratories)-coated plates were used, and T cells were maintained in AIM-V media (Thermo Fisher Scientific) supplemented with 5% human AB serum (Valley Biomedical), containing 100 IU/ml IL-2 (R&D).

Lentiviral gene transduction was performed to overexpress iRFP720 or firefly luciferase in tumor cell lines. To produce lentivirus, 293T/17 cells were transfected with 2 μg transfer plasmid along with 1.6 μg psPAX2 (Addgene) and 0.8 μg pMD2.G using 10 μl Lipofectamine 2000 in 600 μl OPTI-MEM. Two days later, the supernatants were harvested, and transduction was conducted in the presence of 10 μg/ml polybrene (Sigma-Aldrich).

**Flow cytometry**. FACS samples were run on a BD Biosciences LSR-II Flow Cytometer, and data analysis was performed using FlowJo software (TreeStar, Ashland, OR). Forward- and side-scatter gating along with propidium iodide staining was used to differentiate the lymphocyte population and live and dead cells. Surface and intracellular expressions of various markers were determined using the following antibodies from BD Biosciences: CD3, CD4, CD8, CD62L, CD45RA, CD57, PD-1, CD70, CXCR1, and CXCR2. All related antibodies are listed in Supplementary Table 2.

**T-cell proliferation assay**. To evaluate the effects of IL-8 on the modified CAR T-cell proliferation, T cells were labeled with the CellTrace Violet Cell Proliferation Kit according to the manufacturer's protocol on day 0 and stimulated with anti-CD3/anti-CD28 Dynabeads (Thermo Fisher Scientific) with the indicated dose of rhIL-8. Four days later, proliferation was assessed by flow cytometry. All experiments were done in triplicates.

**Luciferase-based cytotoxicity assay**. A total of $2 \times 10^4$ luciferase-expressing U87.Luc target cells were cocultured with various amounts of CAR T cells or control T cells, respectively, as indicated by the effector-to-target (E:T) ratios, for 18 h in black-walled, 96-well flat-bottom plates. The target cells alone were seeded at the same density to determine the spontaneous death relative light units (RLUs), and the target cells were treated with 0.1% Triton-X-100 (Sigma-Aldrich) to measure maximal killing RLU. After 18 h, 100 μl/well of the supernatant was removed, and 100 μl of BrightGlo Reagent (Promega) was added for the luciferase assays. Luminescence was measured by the Cytation 3 Cell Imaging Multi-Mode Reader (BioTek) after 10 min of incubation to allow complete cell lysis. Triplicated wells were averaged, and percent lysis was calculated from the data with the following equation: % specific lysis = 100% × (spontaneous death RLU − test RLU)/(spontaneous death RLU − maximal killing RLU).

**Transwell migration assay**. T cells ($2 \times 10^5$) were placed in the upper chamber (5.0-μm pore size) of a well in a 24-well Transwell plate (Corning), and the lower chamber was filled with an assay buffer (1% BSA/AIM-V) with different concentrations of rhIL-8 (Sigma-Aldrich) or conditioned media from a radiated U87 cell culture. After 1 h, cells from the lower chamber were collected, and the number of cells was counted by flow cytometry with the CountBright Absolute Counting Beads (Thermo Fisher Scientific).

**Murine xenograft tumor model**. To test if radiation could enhance IL-8 expression by glioma in vivo, two orthotopic xenograft mouse models were used. Six- to eight-week-old female NSG-B2m mice (NOD.Cg-B2mtm1UncPrkdcscid Il2rgtm1Wjl/SzJ, the Jackson Laboratory) or NRG mice (NOD.Cg-Rag1tm1Mom Il2rgtm1Wjl/SzJ, the Jackson Laboratory) were implanted with $5 \times 10^4$ U87 glioma cells by i.c. injection and fractionated local radiations were performed at indicated doses. For the adoptive transfer treatment study, NRG mice were implanted with U87.i720 glioma cells by i.c. injection and two fractionated radiations were performed. For the subcutaneous mouse model, $1 \times 10^6$ PANC-1.i720 pancreatic tumor cells or SK-OV-3.i720 ovarian tumor cells were inoculated subcutaneously into the right-hind flank of NRG mice and fractionated local radiations were performed. Seven or eight days after the first radiation, $2 \times 10^6$ CAR T cells were delivered systemically via tail-vein injection.

**Monitoring of tumor growth and T-cell trafficking in vivo**. Isoflurane-anesthetized animals were imaged using the IVIS system (Xenogen, Alameda, CA). A fluorescent signal of the near-infrared protein iRFP720 was detected first for tumor size identification. Then the luminescent signal of CBluc was detected 10–15 min after 150 mg/kg D-luciferin (PerkinElmer) injection intraperitoneally. The average radiant efficiency ([p/s/cm$^2$/sr]/[μW/cm$^2$]) within the ROI (i.e., brain in i.c. model or right-hind flank in s.c. model) was used for quantitative fluorescence analysis. The average radiance (p/s/cm$^2$/sr) within the ROI was used for all quantitative luminescence analyses. Tumor size and T-cell signals were tracked at the time points indicated in the figures.

**Cell isolations**. The harvested tumor was digested with 1 mg/ml collagenase and 1 mg/ml DNase-I digestive enzymes (Collagenase type I-S, Sigma Aldrich. Deoxyribonuclease I, Worthington Biochemical Corp.) at 37 °C for 15 min to acquire tumor single-cell suspensions. The intratumoral CAR T cells were then enriched by gradient centrifugation with 70–30% Percoll (GE Healthcare) at 500 g for 30 min at 22 °C. The fresh spleen was mashed with a 70 μm cell strainer, and the cells underwent red blood cell lysis using Lysing Buffer (BD Biosciences). All cells were washed twice before flow cytometry analysis.

**Statistical methods**. All experiments were performed in duplicate at the very least. For comparison between two groups, an unpaired t test and Mann–Whitney U test were used. For comparisons of three or more groups, the values were analyzed by one-way ANOVA. Survival determined from the time of tumor implantation was analyzed by the Kaplan–Meier method and the log-rank test (*$P < 0.05$, **$P < 0.01$, ***$P < 0.001$).

**Study approvals**. In murine studies, mice were handled in accordance with the University of Florida (UF) animal care policy, and all protocols of the studies were approved by UF's Institutional Animal Care and Use Committee [IACUC Study #201809104 (PI: Huang, J.): T-cell therapy against solid tumors, and IACUC Study #201609208 (PI: Huang, J.): The combination therapies for gliomas]. All constructs used in this report were evaluated and approved by the UF's Division of Environmental Health & Safety, Biological Safety Office [RD-4231(PI: Huang, J.): The role of CD70 in antitumor response]. Human materials were handled according to the federal regulations and approved by the UF's Institutional review board [Study #IRB201400101 (PI: Huang, J.): Testing the function of human immune cells using isolated PBMC from Healthy Donors].

## Data availability

The source data underlying Figs. 1a–e, 2c, e–i, 3d–f, i–k, n–p, 4b, d–f, 5d, e, g, h, j, l, m, 6c–h, k–m, and Supplementary Figs. 1a–f, 3b are provided as a Source Data file. The data that support the findings of this study are available within the article and its Supplementary Information files and from the corresponding author upon reasonable request.

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

## Acknowledgements

This work was supported by the Wells Endowment, The Florida Center for Brain Tumor Research (FCBTR), Accelerate Brain Cancer Cure (ABC²), and UFHCC/Wells BTC Pilot Award. We thank Ginger Moore and Hector Mendez for technical assistance.

## Author contributions

Conception and design: L.J. and J.H. Development of methodology: L.J., H.T., A.K., Y.L., A.Y.H., M.N., K.A.D., A.J.G., L.P.D., D.A.R., W.Z., F.J.B., Q.J.W., J.C.Y., J.L.K., Z.L., D.A.M., and J.H. Analysis and interpretation of data: L.J., H.T., K.D.A., M.N., E.J.S., Z.L., and J.H. Writing, review, and revision of the paper: L.J., A.K., K.A.D., A.J.G., L.P.D., A.Y.H., E.J.S., M.R., and J.H. Study supervision: J.H.

## Additional information

**Competing interests:** The University of Florida has published PCT patent applications related to this project (Publication no. WO 2019/051047, inventors: L.J. and J.H.). Remaining authors declare no competing interests.

