## [Peer Review File · Nature Communications]

Reviewers' Comments:

Reviewer #1:

Remarks to the Author:

The study by Jin et al examine the capacity of CAR T cells directed against the CD70 antigen modified with IL-8 receptors, CXCR1 or CXCR2 (CAR.R1/CAR.R2) to target glioma tumor cells in vitro and in a xenograft tumor model. The authors show in a number of glioma lines that ionizing radiation increases IL-8 production that can potentially promote tumorigenesis. They test the hypothesis that reducing the level of this cytokine within the tumor microenvironment using IL-8R transduced CAR T cells may enhance the trafficking of these adoptively transferred T cells resulting in reducing tumor growth. In chemotaxis assays they showed that IL-8 could enhance the proliferation of anti-CD70 CAR.R1 cells but not CAR.R2 T cells. Furthermore they show that adoptive transfer of both CAR.R1 and CAR.R2 T cells could effectively eradicate establish U87 glioma tumor cells in vivo and induce a memory response against tumor rechallenge. Overall this strategy is novel and may lead to improving the targeting of CAR T cells against solid cancer. However there are a number of several questions that need to be addressed for improving the significance of the study.

1) The authors show that adoptive transfer of CAR.R1 or CAR.R2 T cells can effectively eradicate establish U87 cells compared to control CAR.EGFP T cells. Apart from increased migration and persistence of these cells to the tumor site there is a lack of information on other functional parameters including cytokine release and cytotoxic potential of CAR.R1 and CAR.R2 T cells compared to control CAR T cells. How does the phenotype of these cells compare to control CAR T cells in vivo?

2) The authors demonstrate that CAR.R2 T cells did not proliferate in vitro in response to IL-8 yet these cells responded well in vivo in adoptive transfer experiments. In fact Fig. 5 shows that CAR.R2 T cells were more effective. However, it is not clear why this was the case? There is no experimental data shown to explore the differences the authors observe between in vitro and in vivo experiments for CAR.R2 T cells responding to IL-8.

3) In Figure 4F how representative is this data? Interestingly the authors show lack of function of CAR.EGFP control T cells isolated from U87 tumor cells in terms of cytokine production and high PD-1 expression on these cells. How does this compare to CAR.R1 and CAR.R2 T cells at early time points?

4) Does adoptive transfer of CAR.R1 or CAR.R2 T cells result in decreasing IL-8 levels within the tumor microenvironment following radiation compared to control CAR T cells?

5) In Fig.4K, data for only one mouse receiving CAR.EGFP T cells is shown? This sample size needs expanding for any meaningful conclusions to be reached.

6) In this study, the effects of CAR.R1 and CAR.R2 T cells on one tumor cell line is shown in vivo which is rather limited. The capacity of CAR.R1 and CAR.R2 T cells to mediate anti-tumor effects compared to control CAR T cells in some of the other tumor lines shown to produce IL-8 in response to radiation should be examined to increase the impact of the work.

Reviewer #2:

Remarks to the Author:

The manuscript by Linchun Jin and colleagues aimed to tackle the key obstacles that CAR T-cell therapy is facing in solid tumors by modifying the CD70 CAR with interleukin 8 receptors (IL-8R), CXCR1 and CXCR2, utilizing tumor origin IL-8 secretion. They showed that the IL-8R-modified CAR-transduced T cells markedly improved intratumoral CAR T-cell trafficking/persistence and

provide long-lasting tumor control. Authors from the very beginning (Abstract) underlined the significance of this study in the context of the clinically relevant glioblastoma model, which epitomizes the challenges for the therapy due to its heterogeneity, highly invasive nature, and geographic location. Unfortunately, model used in in vitro and in vivo studies do not address such obstacles.

Specifically:

A. Author's used engineered U87 cell and used them in all in vitro/in vivo experimental testing – (F1d, F3a-c, F4a-m, F5a-j)

1. this model do not provide any heterogeneity

2. this model is not invasive in vivo

3. this model do not address geographic location/ is niche independent

4. there is an ongoing debate whether U87 are what they were thought to be

(<https://www.nature.com/news/venerable-brain-cancer-cell-line-faces-identity-crisis-1.20515>)

The primary cells with subtype characteristic and with diverse phenotype (invasive, nodular like) followed by co-culture/co-injection (in vitro and in vivo) should be used.

B. There is lack of mechanistic studies to explain intricacies of the pathways

C. Using PMBC from healthy donors is not correct model – should be material from patient with match tumor derived cells. The experiments were performed with activated and presumably immunologically functional T cells from healthy human volunteers, it is not known whether this also can be reproduced with dysfunctional and immunologically "exhausted" T cells from human glioblastoma patients – this need to be validated using match tumor cells and T-cells.

D. Analysis of immuno-related secretome of glioblastoma was already published - also using more relevant tissue microdissection ISH and deep seq analysis in tumor anatomic sites regions (Please also see IVY GAP dataset) so data on FS1 are not novel.

E. F2S Authors have shown level of IL18 in different cell line. This experiment has no proper control – does this really increases of expression as authors claim, or is this an increase secretion, or maybe this depends on radiation sensitivity? How is this relevant to in vivo secretion? How authors defined secretion from living/death cells. In situ histochemistry with radiation should be shown using this entire model in vivo. Also other chemokine needs to be analyzed (F1AB).

Reviewer #3:

Remarks to the Author:

This is an interesting study on updating CAR T cell therapy to focus on the target CD70 with an interleukin-8 (IL-8) modification to further enhance the migration of the CAR T cell to tumor.

Introduction

As the study appears to present IL-8 modification as a novel aspect, introducing the rationale and some more background regarding IL-8 itself earlier in the introduction would help the this section flow better. Perhaps, the paragraph starting with "The expansion and persistence of CAR T cells..." can be moved up.

Results

Characteristics of IL-8R-modified CD70 CAR T cells in vitro

Figure 2e: Were there any significant differences in the influence of the different rh-IL8 levels for each construct?

Figure 2g: By "relatively more vigorous cell migration etc.", there was no significance for CAR.R1 over CAR.R2? as the figure itself doesn't indicate whether there is.

The modified CARs enhance T-cell tumor migration and precipitate cures in tumor bearing animals

Are there any images showing relative tumor size prior to the two local radiation doses compared to the time period between Day 11-17 (prior to injection of CAR T cells)?

The anergic/exhausted phenotype of the intratumoral CAR T cells in relapsed settings

Figure 4a: It is interesting looking at the images of luminescence for CAR.R2 compared to CAR.R1. There appears to be widespread luminescence for CAR.R2 whereas in Figure 3, both groups have similar appearances. Does your results or discussion write-up comment on this?

Did your group investigate levels of regulatory T cells or other checkpoint receptor expression on T cells?

Modified CAR T cells cure late-stage tumors and provide long-lasting tumor protection

It would be interesting to see the phenotype profile (whether there is any downregulation of checkpoint receptors etc.) of the CAR.R1 and CAR.R2 T cells after tumor rechallenge in similar experiments as the previous Results sections on exhausted CAR.EGFP T cells.

Discussion

“Delayed treatment and trafficking of unmodified cells precede the formation of a suppressive intratumoral microenvironment with dysfunctional CAR T cells, which we have observed in our xenograft models.” The phrasing is a bit confusing; perhaps replace with “delayed treatment...promote the formation of....”

Since one the key points sounds like early and fast migration of modified CAR T cells is critical for tumor abolishment and avoidance of the development of a suppressive tumor microenvironment, was there any evidence in in vitro studies on the migratory time course of the modified CAR T cells? i.e. if the modified CAR T cells demonstrate a surge in migratory capability in vitro. The closest was Figure 2e, but it was comparing different concentrations of IL-8.

Perhaps, the final two sentences of the discussion section can be swapped in order to emphasize the novelty of utilizing IL-8 production and modification, since previous papers have described the significant of CD70 targeting already.

We believe that this manuscript may be suitable for publication with the aforementioned suggestions.

REVIEWERS' COMMENTS:

Reviewer #1

(Remarks to the Author):

1) The authors show that adoptive transfer of CAR.R1 or CAR.R2 T cells can effectively eradicate establish U87 cells compared to control CAR.EGFP T cells. Apart from increased migration and persistence of these cells to the tumor site, there is a lack of information on other functional parameters including cytokine release and cytotoxic potential of CAR.R1 and CAR.R2 T cells compared to control CAR T cells. How does the phenotype of these cells compare to control CAR T cells in vivo?

Answer:

We provide a comparison of intratumoral CAR T cell phenotype after in vivo treatment (**new Fig. 4**). Compared with the CAR.R1/R2 T cells, very few CAR.EGFP T cells are observed inside the tumor on day 2 post-treatment. Modified CAR T cells also appear to have a more activated phenotype (i.e., more GZMB and less PD-1) than unmodified CAR T cells (2 days post-treatment). The short interval between the start of treatment and complete tumor-regression (4-6 days) after CAR.R1/R2 limits testing of intratumoral CAR T cell phenotype at later time points.

2) The authors demonstrate that CAR.R2 T cells did not proliferate in vitro in response to IL-8 yet these cells responded well in vivo in adoptive transfer experiments. In fact Fig. 5 shows that CAR.R2 T cells were more effective. However, it is not clear why this was the case? There is no experimental data shown to explore the differences the authors observe between in vitro and in vivo experiments for CAR.R2 T cells responding to IL-8.

Answer:

While CAR-R2 is superior to CAR-R1 in vivo, CAR-R1 displayed better chemotaxis in response to IL-8 in vitro. To demonstrate if these conflicting observations would hold true in separate tumor models, we explored CAR-R1 and CAR-R2 activity in distinct ovarian and pancreatic cancer models (newly added section of the **Fig.3g-o**); CAR-R2 remained consistently superior. While the reasons remain unclear, we have amended our discussion that the question requires further experimentation.

3) In Figure 4F how representative is this data? Interestingly the authors show a lack of function of CAR.EGFP control T cells isolated from U87 tumor cells in terms of cytokine production and high PD-1 expression on these cells. How does this compare to CAR.R1 and CAR.R2 T cells at early time points?

Answer:

The result in the original Figure 4F (Fig.5F in the current version) is representative of 3 animals.

We added **new Fig.4** to address your concerns. We uncovered, at early time points post-treatment, a trend towards better phenotype of CAR.R1/R2 (\uparrow GZMB and moderate PD-1 expression), compared with CAR.EGFP transduced T cells *in vivo*.

4) Does adoptive transfer of CAR.R1 or CAR.R2 T cells result in decreasing IL-8 levels within the tumor microenvironment following radiation compared to control CAR T cells?

Answer:

Since these 2 modified CAR T cells decrease IL-8 in *in vitro* co-cultures, we hypothesized that IL-8Rs modified CAR T cells would enhance their *in vivo* trafficking; however, it remains unclear from our studies if *in vivo* IL-8 is decreased post-RT after administration of modified CAR T cells. Our ongoing study will answer this question.

5) In Fig.4K, data for only one mouse receiving CAR. EGFP T cells are shown? This sample size needs expanding for any meaningful conclusions to be reached.

Answer:

Fig. K is representative imaging, and data showed in **Fig. 5i and 5m** (revised) display two mice. Sample sizes should be greater; however, catching an individual relapsed mouse for retreatment is challenging in the GBM model, because mice with tumor recurrence (~50%) showed neurological symptoms that have to be sacrificed before or during the early stage of the retreatment according to the IACUC requirement.

In this study, the effects of CAR.R1 and CAR.R2 T cells on one tumor cell line is shown *in vivo* which is rather limited. The capacity of CAR.R1 and CAR.R2 T cells to mediate anti-tumor effects compared to control CAR T cells in some of the other tumor lines shown to produce IL-8 in response to radiation should be examined to increase the impact of the work.

Answer:

We agree with the reviewer. To broaden the impact of this work, we added two additional tumor models (ovarian and pancreatic cancers, in new **Fig. 3g-o**) and corroborated the results obtained in our original GBM model.

Reviewer #2:

(Remarks to the Author):

A. Autor's used engineered U87 cell and used them in all in vitro/in vivo experimental testing – (F1d, F3a-c, F4a-m, F5a-j)

1. This model does not provide any heterogeneity
2. This model is not invasive in vivo
3. This model does not address geographic location/ is niche independent
4. There is an ongoing debate about whether U87 are what they were thought to be (<https://www.nature.com/news/venerable-brain-cancer-cell-line-faces-identity-crisis-1.20515>)

The primary cells with subtype characteristic and with diverse phenotype (invasive, nodular like) followed by co-culture/co-injection (in vitro and in vivo) should be used.

Answer:

While U87 cells express CD70 (see Figures below), we fully agreed that they might not be representative of GBM. During this ~year-long revision, we have attempted to stratify this criticism by generating orthotopic xenograft models using primary GBM lines. We experimented with three CD70+ primary lines, including two lines used in our previous report (Jin, L et al., Neuro-Oncology, 2018). Unfortunately, none of these lines were successfully grafted. Thus, we decided to expand our findings in other cancer types (ovarian and pancreatic cancer models) that naturally express CD70, and also secrete IL-8 by radiation (**Fig. S1, S3, and the new Fig. 3g-o**). We obtained the same results in these models as we had in our GBM model (**Fig. 3g-o**).

B. There is a lack of mechanistic studies to explain the intricacies of the pathways

Answer:

We agree that mechanistic studies would provide further insights into the work articulated herein, but over the last year, we have devoted our efforts to repeating our

observations in different tumor model systems while expounding on the developmental work regarding this new CAR T cells technology for solid tumors.

C. Using PMBC from healthy donors is not the correct model – should be material from a patient with match tumor-derived cells. The experiments were performed with activated and presumably, immunologically functional T cells from healthy human volunteers; it is not known whether this also can be reproduced with dysfunctional and immunologically "exhausted" T cells from human glioblastoma patients – this need to be validated using match tumor cells and T-cells.

Answer:

This is an excellent point. In the revision, we performed a validation between PBMCs derived from two GBM patients (**new Fig.3g-o and new Fig.4**) versus healthy donors. No differences were observed in *in vitro* and *in vivo* efficacy of CAR-transduced T cells between the PBMCs from two GBM patients versus healthy donors. In addition, based on our phenotyping analysis, we did not see downregulation of CD27 (a memory T cell marker) in GBM patients' blood, compared with the healthy donors (see Figure below).

D. Analysis of immuno-related secretome of glioblastoma was already published - also using more relevant tissue microdissection ISH and deep seq analysis in tumor anatomic sites regions (Please also see IVY GAP dataset) so data on FS1 are not novel.

Answer:

Based on your suggestion, as well as a shift of focus from GBM to multiple tumor models, we removed Fig.S1 from this revision.

E. F2S Authors have shown a level of IL8 in the different cell line. This experiment has no proper control – does this really increases of expression as authors claim, or is this an increase secretion, or maybe this depends on radiation sensitivity? How is this relevant to *in vivo* secretion? How authors defined secretion from living/death cells. *In situ*

histochemistry with radiation should be shown using this entire model *in vivo*. Also other chemokine needs to be analyzed (F1AB).

Answer:

This experiment (**Fig. S2, Fig. S1** in the current version) was based on irradiation on U87 line *in vitro* (Fig.1b, increased IL-8 secretion by ELISA), as well as *in vivo* (Fig.1f, increased secretion of IL-8 from U87 xenograft, measured by IHC from tumor sections). A dose-dependent increase of IL-8 production by irradiation was observed. In addition, most of these cancer types have been shown to be radio-insensitive. Thus, we believe that the enhanced IL-8 secretion was induced by the irradiation. Other chemokines may also be increased by the irradiation, but IL-8 was selected for this study for the new CAR modifications.

Importantly, in this revised version, 3 out of the 6 lines were modeled and tested *in vivo*, and local radiation did not alter tumor growth but enhanced intratumoral migration of modified CAR T cells.

Reviewer #3:

(Remarks to the Author):

Introduction

As the study appears to present IL-8 modification as a novel aspect, introducing the rationale and some more background regarding IL-8 itself earlier in the introduction would help this section flow better. Perhaps, the paragraph starting with “The expansion and persistence of CAR T cells...” can be moved up.

Answer:

We re-wrote most of the introduction according to your suggestion and included the background of IL-8 in the section.

Results

Characteristics of IL-8R-modified CD70 CAR T cells *in vitro*

Figure 2e: Were there any significant differences in the influence of the different rh-IL8 levels for each construct?

Answer:

To address this concern, we added additional info to the statistical analysis shown in **Fig.2e**.

Figure 2g: By “relatively more vigorous cell migration etc.”, there was no significance for CAR.R1 over CAR.R2? as the figure itself does not indicate whether there is.

Answer:

We added additional information regarding statistical analysis to this figure.

The modified CARs enhance T-cell tumor migration and precipitate cures in tumor-bearing animals

Are there any images showing relative tumor size prior to the two local radiation doses compared to the time period between Day 11-17 (prior to injection of CAR T cells)?

Answer:

We did not measure the tumor volumes before and after the radiation since U87 is a radioresistant cell line, and also, all groups were treated with the same schedule and dose of irradiation.

The anergic/exhausted phenotype of the intratumoral CAR T cells in relapsed settings

Figure 4a: It is interesting looking at the images of luminescence for CAR.R2 compared to CAR. R1. There appears to be widespread luminescence for CAR.R2 whereas in Figure 3, both groups have similar appearances. Does your results or discussion write-up comment on this?

Answer:

We have shown CAR-R2 T cells were superior in *in vivo* persistence and antitumor efficacy, compared to CAR.R1 in GBM, ovarian and pancreatic cancer models (**Fig. 3 and Fig. 6**). The mechanism is unclear; we included comments in the discussion.

Did your group investigate levels of regulatory T cells or other checkpoint receptor expression on T cells?

Answer:

We added a new **Fig. 4c, e**, suggesting a trend toward to that the CAR. EGFP T cells express higher PD-1 than CAR.R2 T cells. In addition, we are currently investigating these questions in syngeneic mouse models using the mouse version of CD70CAR.

Modified CAR T cells cure late-stage tumors and provide long-lasting tumor protection

It would be interesting to see the phenotype profile (whether there is any downregulation of checkpoint receptors etc.) of the CAR.R1 and CAR.R2 T cells after tumor rechallenge in similar experiments as the previous Results sections on exhausted CAR.EGFP T cells.

Answer:

Due to the high potency of the CAR-R1 and CAR-R2 T cells, a short interval exists between the start of treatment and complete tumor regression making it difficult to procure enough T cells from the tumors. However, we did investigate the phenotypes, including PD-1 expression between the modified and unmodified CAR T cells in the

“primary” tumors at the early time point (**new Fig.4**). There was a trend toward lower PD-1 levels in modified versus unmodified CAR T cells.

Discussion

“Delayed treatment and trafficking of unmodified cells precede the formation of a suppressive intratumoral microenvironment with dysfunctional CAR T cells, which we have observed in our xenograft models.” The phrasing is a bit confusing; perhaps replace with “delayed treatment...promote the formation of...”

Answer:

We clarified this sentence as you suggested.

Since one the key points sounds like early and fast migration of modified CAR T cells is critical for tumor abolishment and avoidance of the development of a suppressive tumor microenvironment, was there any evidence in *in vitro* studies on the migratory time course of the modified CAR T cells? i.e. if the modified CAR T cells demonstrate a surge in migratory capability *in vitro*. The closest was Figure 2e, but it was comparing different concentrations of IL-8.

Answer:

We only tested a one-time point and found differential migration outcomes *in vitro*. In addition to the *in vitro* assays, we show the *in vivo* migrative kinetics in these CAR T cells (**Fig. 6a, c**).

Perhaps, the final two sentences of the discussion section can be swapped in order to emphasize the novelty of utilizing IL-8 production and modification, since previous papers have described the significance of CD70 targeting already.

Answer:

Thank you for your suggestion; we modified the discussion section substantially. As you suggested, we have emphasized the clinical implications and further investigations of this study in the last two paragraphs.

Reviewers' Comments:

Reviewer #1:

Remarks to the Author:

The authors have addressed some of the points raised however have neglected other points.

A key mechanism underlying the increased effect of the modified IL-8R CAR's following radiation should be a reduction in IL-8 levels within the tumor microenvironment allowing better infiltration and function of the T cells. The authors have not attempted to address this question.

The authors make the comment that the modified CARs have a more activated phenotype yet they observed a decrease in PD-1 expression. It would be expected that PD1 would be increased. Can the authors comment further on this observation?

The authors have not shown any conclusive data on why CAR.R2 modified T cells perform better than CAR.R1 T cells in vivo yet CAR.R1 T cells display better in vitro functional characteristics in vitro ie.stronger chemotaxis towards IL-8

Reviewer #2:

Remarks to the Author:

I am mostly satisfied by the author's efforts to address criticisms. Still use of U87 is barely acceptable at this level.

Serious limitations of this model should be disclosed.

Reviewer #3:

Remarks to the Author:

Comments have been addressed.

REVIEWERS' COMMENTS:

REVIEWER #1 (REMARKS TO THE AUTHOR):

The authors have addressed some of the points raised however have neglected other points. A key mechanism underlying the increased effect of the modified IL-8R CAR's following radiation should be a reduction in IL-8 levels within the tumor microenvironment allowing better infiltration and function of the T cells. The authors have not attempted to address this question.

Answer:

We agree with you.

We hypothesize that the IL-8R modified CAR T cells potentially act as a sink to neutralize or remove the IL-8 induced tumor immunosuppression, which may help to improve the efficacy. However, more evidence is needed to draw this conclusion. We have discussed our hypothesis in the discussion (page 13, lines 10-18).

The authors make the comment that the modified CARs have a more activated phenotype yet they observed a decrease in PD-1 expression. It would be expected that PD-1 would be increased. Can the authors comment further on this observation?

Answer:

No significant decrease in PD-1 expression was found on the mod-CAR T cells compared with the un-mod CAR T cells (only a trend of decrease, Fig. 4e).

The increased expression of GZMB and superior in vivo activity of the mod-CAR T cells led us to conclude that the mod-CAR T cells present a more activated phenotype. Although PD-1 is also can be expressed on activated T cells, but it is an inhibitory molecule related to T cell dysfunction in cancer. The analysis of intratumoral CAR T cells isolated from the relapsed tumors (and spleen) showed that the PD-1 was significantly elevated compared with the baseline, and these CAR T cells were unresponsive to the tumor target (Fig. 5g-h). These data and previously reported result by other groups suggest that PD-1 down-regulation on the CAR T cells is associated with activated phenotype.

The authors have not shown any conclusive data on why CAR.R2 modified T cells perform better than CAR.R1 T cells in vivo yet CAR.R1 T cells display better in vitro functional characteristics in vitro ie. stronger chemotaxis towards IL-8.

Answer:

To date, the molecular basis for the differential regulation of the IL-8 receptors remains unclear, however, modifying CAR using these receptors to enhance T cell trafficking to tumor, and to improve antitumor efficacy is the focus of this study. Certainly, further studies on unique immunological functions between these two modifications may answer the question specifically. We have mentioned a potential study in the future in the discussion section.

REVIEWER #2 (REMARKS TO THE AUTHOR):

I am mostly satisfied by the author's efforts to address criticisms. Still use of U87 is barely acceptable at this level.

Serious limitations of this model should be disclosed.

Answer:

We agree with reviewer that U87 GBM line has limitations. However, the gene analysis suggests that it is a GBM, although some of its characteristics may have shifted from the original tumor. Here is a paragraph from the report you mentioned ([doi:10.1038/nature.2016.20515](https://doi.org/10.1038/nature.2016.20515)):

A comparison of gene-expression profiles conducted by Westermarck's team suggests that the ATCC cell line probably came from a brain tumour. "It's bad news that it's not what it should be," Westermarck says, "but it's good news that it's probably a glioblastoma." This means that studies of U87 still reflect brain-cancer biology and don't need to be tossed out, he adds.

We have result of STR matching analysis showing that the U87 line used in our report is 100% matches the ATCC line (see below Table).

Importantly, we added two additional tumor models which confirmed our findings in the U87 GBM model.

Result of STR matching analysis by your data.													
- DSMZ Profile Database -													
A graphical presentation is shown at the bottom of this page.													
EV	Cell No.	Cell name	Locus names										Figures
			D5S818	D13S317	D7S820	D16S539	VWA	TH01	AM	TPOX	CSF1PO		
		Query (Your Cell)	11.12	8.11	8.9	12.12	15.17	9.3.9.3	X.X	8.8	10.11		
1.00(36/36)	HTB-14	U-87MG	11.12	8.11	8.9	12.12	15.17	9.3.9.3	X.X	8.8	10.11	-	-
0.72(26/36)	731	CAKL-1	11.12	11.11	8.12	12.12	15.17	6.8	X.X	8.11	10.11	-	-
0.72(26/36)	749	U-CH2	10.11	11.11	8.12	12.12	17.17	9.3.9.3	X.X	8.8	11.12	-	-
0.72(26/36)	CRL-5842	NCI-H774 [H774]	11.11	8.8	9.11	12.12	15.17	6.9.3	X.X	8.8	10.10	-	-
0.72(26/36)	CRL-5910	NCI-H1994 [H1994]	10.11	11.11	9.11	12.12	15.19	7.9.3	X.X	8.8	10.11	-	-

REVIEWER #3 (REMARKS TO THE AUTHOR):

Comments have been addressed.

** See Nature Research's author and referees' website at www.nature.com/authors for information about policies, services and author benefits

This email has been sent through the Springer Nature Tracking System NY-610A-NPG&MTS